# Pharmacological Inhibition of Microglial Proliferation Supports Blood–Brain Barrier Integrity in Experimental Autoimmune Encephalomyelitis

**DOI:** 10.3390/cells14060414

**Published:** 2025-03-12

**Authors:** Nozha Borjini, Mercedes Fernandez, Luciana Giardino, Lydia Sorokin, Laura Calzà

**Affiliations:** 1Research & Development, Chiesi Farmaceutici S.p.A, via Palermo 26/A, 43100 Parma, Italy; 2IRET Foundation, via Tolara di Sopra 41/E, Ozzano Emilia, 40064 Bologna, Italy; mercedes.fernandez@unife.it; 3Department de Fisiología Médica y Biofísica, Instituto de Biomedicina de Sevilla (IBiS, Hospital Universitario Virgen del Rocío/CSIC/Universidad de Sevilla), 41013 Sevilla, Spain; 4Facultad de Medicina and CIBERNED ISCIII, 41013 Sevilla, Spain; 5Health Science and Technologies Interdepartmental Center for Industrial Research (HST-ICIR), University of Bologna, via Tolara di Sopra 41/E, Ozzano Emilia, 40064 Bologna, Italy; luciana.giardino@unibo.it (L.G.); laura.calza@unibo.it (L.C.); 6Department of Veterinary Medical Sciences, University of Bologna, via Tolara di Sopra 50, Ozzano Emilia (BO), 40064 Bologna, Italy; 7Institute of Physiological Chemistry and Pathobiochemistry, University of Muenster, 48149 Muenster, Germany; sorokin@uni-muenster.de; 8Cells-in-Motion Interfaculty Centre (CIMIC), University of Muenster, 48149 Muenster, Germany; 9Department of Pharmacy and Biotechnology, University of Bologna, via Tolara di Sopra 41/E, Ozzano Emilia, 40064 Bologna, Italy

**Keywords:** microglia, blood–brain barrier, experimental autoimmune encephalomyelitis, multiple sclerosis, GW2580

## Abstract

Blood–brain barrier dysfunction (BBB) is a primary characteristic of experimental autoimmune encephalomyelitis (EAE), an experimental model of multiple sclerosis (MS). We have previously shown that blocking microglial proliferation using GW2580, a selective inhibitor of CSF1R (Colony stimulating factor 1 receptor), reduced disease progression and severity and prevented the relapse phase. However, whether this was due to effects of GW2580 on the functional integrity of the BBB was not determined. Therefore, here, we examine BBB properties in rats during EAE under GW2580 treatment. Our data suggest that blocking early microglial proliferation through selective targeting of CSF1R signaling has a therapeutic effect in EAE by protecting BBB integrity and reducing peripheral immune cell infiltration. Taken together, our results identify a novel mechanism underlying the effects of GW2580, which could offer a novel therapy for MS.

## 1. Introduction

Multiple sclerosis (MS) is an inflammatory demyelinating disease of the central nervous system (CNS) [1]. The main pathological features of MS plaques are blood–brain barrier (BBB) disruption [2], perivascular inflammation [3,4], demyelination, and axonal damage [5]. The BBB is central to normal brain physiology and protection of the cognitive functions of the brain. By constituting the interface between the CNS parenchyma and peripheral compartments, the BBB regulates the entry and exit of ions, nutrients, macromolecules, and energy metabolites [6]. Disorders within the CNS parenchyma can trigger BBB dysfunction and neuroinflammation, thus contributing to the pathogenesis of CNS disorders [7]. It has been shown that in experimental autoimmune encephalomyelitis (EAE), BBB integrity is compromised by focal activity of matrix metalloproteinase (MMP) [8,9,10,11], a zinc-dependent protease, the functions of which in EAE include the modulation of inflammatory processes, through the activation or inactivation of cytokines [10,12], and the cleavage of myelin proteins. Focal activation of the gelatinases, MMP-2 and MMP-9, at the parenchymal border can compromise BBB integrity by selective cleavage of cell-matrix receptors such as dystroglycan and tight junction (TJ) molecules but also through potentiation of chemotactic signals [8,10,13,14,15], facilitating leukocyte migration out of the perivascular cuff and into the CNS parenchyma.

Microglia, the CNS’s resident immune cells, are the main executors of inflammation, which they produce when responding rapidly to exogenous stimuli during CNS injury or infection. Alterations in BBB integrity involve a dynamic response mediated by all components of the neurovascular unit (NVU), and ongoing research suggests that microglial interaction with different components of the NVU is critical for the response to CNS injury [7,16,17]. Indeed, under inflammatory conditions, especially in MS, it has been reported that microglial extensions can directly contact the glial and endothelial basement membranes throughout the cerebral vascular tree (including arteries, capillaries, and veins) [17,18], which renders them ideally positioned to respond quickly to exogenous stimuli from the vessel lumen. Thus, studies have focused on strategies for modulating microglial activity, and it has been shown depleting the microglia offers a promising strategy for eliminating their disease-promoting effects. Indeed, microglial depletion using the pharmacological receptor tyrosine kinase inhibitor of CSF1R (Colony stimulating factor 1 receptor, also known as cFMS) and related kinases, PLX3397 or PLX5622, in EAE (reviewed in [19]) and in Alzheimer’s disease models [20,21,22] reduces neuroinflammation and supports remyelination and recovery [23]. However, recent data from a mouse model of Parkinson’s disease demonstrate that the ablation of microglia can also lead to increased neurotoxicity and behavioral deficits, which were induced when delivering acute treatment with a dopaminergic neurotoxicant [22,24], suggesting both positive and negative effects of microglia depending on the CNS pathology. In addition to having direct effects on neural cells, microglia may also affect the integrity of the BBB, as microglial processes release trophic factors that have been reported to accelerate endothelial cell closure following injury [25].

GW2580 is an orally available specific inhibitor of the tyrosine kinase activity of CSF1R and, to a lesser extent, other related kinases such as FLT3, CD135, c-Kit, and CD117 [26]. This molecule does not deplete microglia but selectively inhibits the proliferation of microglial stem cells within the CNS [27]. In a prion disease mouse model, GW2580 treatment slowed disease progression, reduced neuronal damage [28], and increased neurogenesis [29]. In a mouse model of amyotrophic lateral sclerosis, GW2580 treatment reduced microglial proliferation, leading to an increased lifespan and better motoneuron preservation [30]. Chronic GW2580 treatment, in a mouse model of Alzheimer’s disease, shifted the microglial response towards an anti-inflammatory phenotype and improved cognitive function [31]. In a chemically induced mouse model of MS, GW2580 administration ameliorated disease progression and decreased peripheral immune cell infiltration and inflammatory cytokine expression [32]. Our previous data revealed that treatment with GW2580 decreases EAE clinical severity and prevents the relapse phase [33]. However, they did not reveal how that effect is mediated. To address this question, we examined CNS tissue (spinal cord and cerebellum) from EAE rats with and without GW2580 treatment. Our results demonstrated that GW2580 treatment reduced microgliosis and T-cell infiltration. Furthermore, we demonstrate that GW2580 treatment correlated with reduced MMP-2/9 activities and downregulation of adhesion molecule and inflammatory mediator expression.

## 2. Materials and Methods

### 2.1. Animals

Female Dark Agouti rats (DA) (Harlan, Milan, Italy) were used in this study; they were given ad libitum food and water and housed/caged in a 12 h light/dark cycle. All animal protocols were carried out in accordance with European Community Council Directives (86/609/EEC), approved by the intramural ethical committee for animal experimentation of Bologna University and the Ministry of Health (n° 158/2013-B, n° 607/2016-PR), and complied with the guidelines published in the NIH Guide for the Care and Use of Laboratory Animals [34].

### 2.2. EAE Induction and GW2580 Treatment

Three-month-old female Dark Agouti rats (149–168 g body weight) were immunized with medium containing 0.15 mg/mL guinea pig spinal cord extract in complete Freund’s adjuvant (CFA, Sigma, Saint Louse, MO, USA), 50% *v*/*v*, to which 5 mg/mL heat-inactivated Mycobacterium tuberculosis (Difco H37Ra, DB, Milan, Italy) was added. Immunization was performed by injecting 100 μL in both hind leg foot pads. GW2580 (LC Laboratories, Boston, MA, USA) was dissolved in 0.5% hydroxypropylmethylcellulose and 0.1% Tween 80 (vehicle) [26,35]. Rats were fed daily from day -1 to day 11 after immunization by oral gavage using flexible plastic feeding tubes FTP-15-78-50 (Instech Laboratories, Netherlands) with GW2580 at 40 mg/kg (EAE+GW2580, N = 6) or an equal volume of vehicle (EAE+vehicle, N = 18). The dose of GW2580 was chosen based on prior studies demonstrating its effectiveness in suppressing microglial proliferation while minimizing off-target effects on related kinases [35,36]. Rats were divided into 3 equal groups: EAE day 8, EAE day 11, and EAE day 18 (Figure 1A). Rats were examined daily for clinical signs of EAE, according to the following semi-quantitative scores for neurological disability: 0 = no signs, 1 = loss of tail tone, 2 = mono or bilateral weakness of hind legs or middle ataxia, 3 = ataxia or paralysis, 4 = severe hind leg paralysis, 5 = severe hind leg paralysis and urinary incontinence [37]. Depending on the degree of the animals’ disability, wet food was included inside the cages to facilitate feeding. EAE rats (EAE+Vehicle, N = 6/group) were sacrificed at 8 (EAE day 8), 11 (EAE day 11), and 18 days post-immunization (DPI) (EAE day 18). EAE+GW2580 rats were sacrificed at 18 DPI, the last day of the experiment (EAE day18+GW2580). Same-aged healthy rats (N = 6) were sacrificed and used as controls (CTRL).

### 2.3. Spinal Cord mRNA Analysis

Total RNA was prepared from the spinal cords of the different groups using QIAzol Reagent, cleaned with an RNeasy Mini kit (Qiagen; Milan, Italy), and eluted in RNase-free water, and the purity and concentration were evaluated by spectrophotometry using a NanoDrop ND-2000 (ThermoScientific, Milan, Italy). cDNA synthesis was performed using an RT^2^ First Strand Kit according to the manufacturer’s instructions. Real-time PCR amplification was performed using the Stratagene Mx3005P multiplex quantitative PCR system (Agilent Technologies, Milan, Italy). Investigation of the expression of genes involved in inflammation, cell adhesion, and inflammatory cell infiltration was carried out using RT2 SYBR Green qPCR Mastermix (Qiagen). The raw data obtained were uploaded to GeneGlobe Data Analysis software (SABiosciences, Qiagen ((accessed on February 2019)) and used for analyses. Relative quantification of the mRNA expression was performed using the comparative cycle threshold (CT) method and expressed as the Log 2-fold change in expression. The fold change (2^−ΔΔCt^) was the normalized gene expression (2^−ΔCt^) in the test sample divided by the normalized gene expression (2^−ΔCt^) in the control sample.

### 2.4. Histology

To analyze perivascular cuffs in the cerebellum, tissues were cut into 10 μm sections on a cryostat (Leica CM1950, Walldorf, Germany), and stained with hematoxylin and eosin (H&E) according to standard procedures. Images were captured using a Zeiss LSM800 confocal microscope and analyzed using ImageJ (Version 1.54d).

### 2.5. Immunohistochemistry and Quantification

Cerebellum tissues were fixed in 1.5% paraformaldehyde saturated aqueous solution in 0.1 M Sörensen buffer, pH 7, for 4 h, placed in 30% sucrose solution, and then embedded in Tissue-Tek^®^ O.C.T.™ (Sakura Finetek Europe, Alphen van den Rijn, The Netherlands). Tissues were frozen and kept at −80 °C until processed. Sections (10 μm) were cut on a cryostat (Leica CM1950, Walldorf, Germany), hydrated in PBS, and incubated for 1 h in pre-absorption solution (PBS-0.5% Triton X-100, 1% BSA), followed by overnight incubation at 4 °C with the primary antibodies diluted in the pre-absorption solution. The antibodies and dilutions used in our immunofluorescence studies are detailed in Appendix A. After PBS washes, sections were incubated at 37 °C for 2 h with secondary antibodies diluted in the pre-absorption solution. Sections were then rinsed in PBS and mounted in Elvanol. Control sections were incubated with secondary antibodies only and processed in parallel. Sections were examined using a Zeiss AxioImager equipped with epifluorescent optics or a Zeiss LSM 700 confocal microscope and documented using a Hamamatsu ORCA-ER camera. Images were captured using a Zeiss LSM800 confocal microscope and analyzed using Volocity 5.1 software (Quorum Technologies, Puslinch, ON, Canada) and ImageJ. Three sections from a minimum of four mice were used for each analysis.

### 2.6. Gelatin Gel Zymography

Gelatin gel zymography was performed as described previously [8]. Cerebellum tissue lysates were used. Prepurification of tissue lysates was carried out by incubation with gelatin-Sepharose 4B for 20 min. Equal amounts of total protein were mixed with sample buffer without reducing agent and run on a 4% stacking and 12% separating SDS-polyacrylamide gel, containing 1 mg/mL gelatin to detect pro- and activated forms of MMP-2 and MMP-9. After electrophoresis, gels were washed with 2.5% Triton X-100 in dH20 for 30 min at room temperature to remove SDS and to renature MMP-2 and MMP-9. Gels were then incubated in developing buffer (50 mM Tris-HCL, 0.2 M NaCl, 5 mM CaCl_2_, 0.02% NP-40, pH 7.9) for an optimized length of time (overnight) at 37 °C to permit gelatin cleavage by the renatured MMPs. After overnight incubation, gels were fixed in acetic acid:ethanol:dH_2_O [10:50:40] for 30 min and washed in acetic acid:methanol:dH_2_O [10:50:40] mixture for 30 min. Subsequently, the zymogram was stained with Coomassie Brilliant Blue and destained in acetic acid:methanol:dH_2_O [10:50:40] solution at room temperature for 1 h. Human recombinant MMP-9 and mouse recombinant MMP-2 were used as controls. This permitted a comparison of the relative proportions of pro–MMP-2/9 and activated MMP-2/9 in samples from healthy controls, EAE+Vehicle, at different time points and EAE rats treated with GW2580.

### 2.7. Functional Enrichment Analysis

Gene Ontology (GO) and Kyoto Encyclopedia of Genes and Genomes (KEGG) pathway enrichment analyses were conducted using Enrichr (http://amp.pharm.mssm.edu/Enrichr/ (accessed on 26 May 2020)). GO Biological Process 2018 was selected. Enrichr is a web-based tool that allows the evaluation of annotations with its extensive gene set libraries [38].

### 2.8. Statistical Analysis

Two-way ANOVA followed by a Bonferroni post-test or Tukey’s multiple-comparison tests were used. Data are presented as means ± standard error of the mean, and significance was set at *p*  ≤  0.05. All statistical analyses were performed using GraphPad Prism 8.0 (GraphPad Software).

For the normalization of gene expression on the RT2 PCR Profiler Array, five housekeeping genes—ribosomal protein, large, P1; hypoxanthine phosphoribosyltransferase 1; ribosomal protein L13A; lactate dehydrogenase A; and β-actin were used. The CT was determined for each sample and normalized to the average CT of the five housekeeping genes. A comparative CT method was used to calculate relative gene expression. Data are represented as Log2 fold change relative to the control. The *p*-values were calculated on the basis of Student’s t test of the replicate 2^−ΔCt^ values for each gene in the control group and treatment groups, and *p*-values less than 0.05 were considered significant.

For functional enrichment analysis, the significant terms and pathways were selected with the threshold of an adjusted *p*-value <  0.05 computed with the Fisher exact test.

## 3. Results

### 3.1. GW2580 Decreased EAE Clinical Severity and Prevents the Relapse Phase

The tyrosine kinase activity of CSF1R was inhibited by oral administration of GW2580, commencing 1 day before and 11 days after EAE induction (Figure 1A). The clinical profiles of the disease in the EAE+Vehicle and EAE+GW2580 groups are shown in Figure 1B. While clinical signs of neurological disabilities started at 6–7 days post immunization (DPI) in the EAE+Vehicle group, a delay in the appearance of the neurological disability was observed in the EAE+GW2580 group. A maximum score of 5 was reached by 11–12 DPI in the control group, corresponding to the acute phase, while a significant reduction in the severity of the disease was measured in the EAE+GW2580 group, with a maximum score of 2.8 (*p* < 0.001). Remarkably, EAE+GW2580 animals did not show any relapse phase (*p* < 0.001) and all rats had recovered by the last day of the experiment (18 DPI) (*p* < 0.001).

### 3.2. Rats Treated with GW2580 Have Enlarged but Reduced Number of Penetrated Perivascular Cuffs in EAE

Inflammatory perivascular cuffs, defined as the accumulation of cells in the perivascular space between the endothelial and the parenchymal basement membranes, were detected by H&E staining to mark leukocyte accumulations and by staining for pan-laminin to demarcate the two basement membranes [39]. Inflammatory perivascular cuffs were not detected in the CNS of control naïve rats. However, at the onset of clinical signs of EAE (EAE day 8), perivascular cuffs were detected in the white matter of the cerebellum, with little infiltration of immune cells into the CNS parenchyma (intact cuffs) (Figure 1C–E). As the disease progressed to peak clinical severity (EAE day 11), inflammatory perivascular cuffs became more numerous and the infiltration of peripheral cells into the parenchyma of the cerebellar white matter became readily detectable by both H&E and immunofluorescence staining (Figure 1C,E). At this stage, the majority of the perivascular cuffs were penetrated by leukocytes (broken cuffs) (Figure 1D). Interestingly, the selective inhibition of CSF1R in EAE with GW2580 (EAE+GW2580 day 18) did not reduce the numbers of perivascular cuffs, but rather significantly reduced the number of perivascular cuffs that were penetrated by leukocytes (Figure 1C,D) and reduced leukocyte infiltration into the cerebellum parenchyma (Figure 1D,E). Stereological analyses of immunofluorescence staining revealed significantly enlarged cuffs of EAE rats treated with GW2580 (EAE+GW2580 day 18) compared to EAE day 18 (Figure 1F).

### 3.3. Blocking CSF1R Decreased Cerebellum Glial Activation

To evaluate the microglial activation, immunofluorescence staining for Iba-1 intensity and density was performed on the cerebellum of control, EAE animals (8, 11, and 18 DPI), and EAE+GW2580 at 18 DPI; sections were double-stained for GFAP to mark astrocytes, and thereby the cuff border, and DAPI to mark all nuclei. Representative images are shown in Figure 2. High-intensity staining for Iba-1 was interpreted as microglial activation and was minimal at 8 DPI, where mainly intact perivascular cuffs were observed and perivascular GFAP+ astrocytic endfeet perfectly surrounded blood vessels (Figure 2A, first panel). Iba-1 immunofluorescence revealed an amoeboid form of resident microglia and perivascular macrophages at 11 DPI and 18 DPI (Figure 2A, second and third panels, Figure 2B,C), suggesting activation. A significant decrease in GFAP density surrounding vessels was observed in the acute and relapse phases, suggesting a loss or retraction of astrocytes’ endfeet. Inhibition of CSF1R (EAE+GW2580 t18) resulted in a significant reduction in microglial activation and density to levels comparable to those measured in non-treated EAE animals at 8 DPI. This correlated with the significantly increased astrocytic cell density (Figure 2A).

### 3.4. Altered Expression of BBB-Related Genes in Spinal Cord During Early EAE

To define the mode of GW2580 action, real-time PCR (qPCR) was employed to investigate the expression of 41 genes coding for proteins and enzymes known to contribute to the maintenance of the BBB and/or its penetration by immune cells. Our analyses were performed on the spinal cord samples. The reason for using spinal cords (SC) for these analyses was the large numbers of cuffs/area compared to the cerebellum. The complete list of genes investigated is presented in Appendix A. The results are presented in a clustergram; non-supervised hierarchical clustering was performed to generate a heat map with dendrograms indicating co-regulated genes across groups, and the criteria for significance are reported in the table of the magnitude of gene expression (Figure 3A). Of the 41 BBB genes tested, an early upregulation of most of the genes was observed as soon as symptoms appeared (8 DPI) and before the peak of the disease (11 DPI). Several endothelial cell adhesion genes, such as VCAM-1 and ICAM-1, were upregulated in the SC at 8 DPI. Most of the chemokines, such as CXCL-11, CCL-12, and CXCL-10, were also upregulated starting from 8 DPI (45.79, 14.22, and 11.13 Log 2-fold change). Our data revealed an upregulation starting from 8 DPI of MMP-9, with 3.2 Log 2-fold change. Most of these changes were not detected in SC from rats treated with GW2580 (Figure 3A and Appendix A). Gene ontology enrichment analysis revealed a high enrichment score of genes linked to cell adhesion, immune activation, and inflammation in EAE on day 18 compared to the EAE day 18+GW2580 group (Figure 3B).

### 3.5. GW2580 Attenuates T-Cell Infiltration into EAE

As T cells are the disease-inducing immune cells in EAE, CD3-immunofluorescence was employed to detect the extent of infiltrated T cells. Few CD3+ cells were found within the CNS parenchyma at 8 DPI (Figure 4A, first panel, Figure 4B), with increasing numbers at EAE day 11 and day 18 and the occurrence of CD3+ cells between astrocyte endfeet and within the CNS parenchyma (Figure 4A, second and third panels, Figure 4B). When compared to EAE t18, the treatment with GW2580 seems to maintain the perivascular astrocytic endfeet and significantly attenuate T-cell infiltration (Figure 4A,B).

### 3.6. GW2580 Treatment Correlates with Reduced MMP-2/9 Activity

Matrix metalloproteinases (MMPs), particularly MMP-2 and MMP-9, are known to degrade components of the BBB and promote leukocyte infiltration into the CNS [10,40]. To investigate whether CSF1R inhibition affects MMP activity, we performed gelatin zymography on cerebellum lysates from control, untreated EAE, and GW2580-treated EAE rats at different disease stages (Figure 4C,D). In untreated EAE rats, we observed a progressive increase in MMP-9 activity starting on day 8 post-immunization, with the highest levels detected on day 18. Both pro-MMP-9 (~92 kDa) and its activated form (~82 kDa) were detected, with a significant increase in the ratio of activated to pro-MMP-9 on day 18 (Figure 4C,D). In contrast, GW2580 treatment markedly reduced MMP-9 activation on day 18, with the activated-to-pro-MMP-9 ratio significantly lower than in untreated EAE rats (*p* < 0.01, two-way ANOVA). MMP-2 was constitutively expressed in all groups, with no significant differences in activation across disease stages or treatment conditions. This suggests that GW2580 specifically impacts MMP-9, which has been implicated in the cleavage of astrocytic endfeet proteins and the promotion of leukocyte migration into the CNS parenchyma [8,41]. The reduction in MMP-9 activation in GW2580-treated animals correlated with fewer CD3+ T cells infiltrating the CNS parenchyma, as shown by immunofluorescence (Figure 4). These findings indicate that CSF1R inhibition may preserve BBB integrity, at least in part, by preventing MMP-9-mediated degradation of the parenchymal barrier, thereby limiting immune cell access to the CNS.

## 4. Discussion

Multiple sclerosis (MS) lesions have been classified into several patterns on the basis of demyelination, as well as the nature and persistence of inflammatory responses, which contribute to severe neuronal degeneration [42,43,44,45]. Despite the heterogeneity, alterations in BBB permeability have long been considered a key initiating factor in MS and EAE [46]. Previously, we have demonstrated that oral administration of GW2580, which is a selective inhibitor of CSF1R, an integral tyrosine kinase transmembrane receptor expressed by microglia under normal conditions, decreased EAE clinical severity and prevented the relapse phase. This highlighted the role of CSF1-CSF1R signaling in microgliosis and inflammation in MS [33]. The current study builds on these findings, showing that GW2580 primarily affects leukocyte migration out of perivascular cuffs into the CNS parenchyma rather than their initial penetration of the endothelial barrier. The unchanged number of perivascular cuffs in GW2580-treated rats suggests that immune cell priming, endothelial adhesion, and basement membrane penetration remain unaffected, whereas immune cell reactivation within the cuffs may be impacted.

Perivascular cuffs, consisting of leukocytes, accumulate around post-capillary venules before infiltrating the CNS parenchyma. These structures are commonly observed in MS patients and EAE [10,15]. Notably, GW2580 attenuated T-cell infiltration into the CNS parenchyma. When administered at 10 days post-immunization, it reduced peripheral macrophage numbers and decreased inflammatory foci within the CNS [32]. This effect likely stems from the inhibition of autocrine signaling in inflammatory microglia and macrophages, which reduces neuroinflammatory mediators [47,48,49]. These findings suggest that targeting tyrosine kinase receptors may enhance BBB integrity and mitigate disease progression [32,50].

Leukocyte transmigration across post-capillary venules into the CNS follows distinct steps [51,52,53]. Initially, leukocytes slow within the blood vessel lumen through integrin α4β1 interactions with endothelial VCAM-1 [14,39,51]. Subsequent integrin αvβ2 (VLA-1)–ICAM-1 binding facilitates leukocyte arrest, possibly in response to endothelial chemokine signaling [54,55]. Once adhered, leukocytes traverse the endothelial barrier and basement membrane, accumulating in the perivascular space to form inflammatory cuffs [8,39]. Our mRNA analysis (spinal cord tissue) revealed the upregulation of most genes related to the BBB structure and function in EAE rats at symptom onset (8 DPI). The BBB’s integrity relies on components such as tight junction-sealed endothelial cells, pericytes, astrocytic endfeet [56,57], and the endothelial and parenchymal basement membranes [41]. Under GW2580 treatment, perivascular cuffs remained intact, with astrocytic endfeet maintaining a normal coverage of blood vessels. Gene ontology enrichment analysis at 18 DPI showed a strong enrichment of genes linked to cell adhesion, immune activation, and inflammation; GW2580 reversed most of these gene expression changes, leading to a significantly lower enrichment score.

Focal activation of the gelatinases, MMP-2 and MMP-9, at the parenchymal border can compromises BBB integrity by cleaving cell-matrix receptors and amplifying chemotactic signals, facilitating leukocyte migration from perivascular cuffs into the CNS. Gel zymography revealed that CSF1R inhibition via GW2580 lowered activated MMP-9 levels at 18 DPI, correlating with reduced CD3+ cell infiltration into CNS tissue. While matrix metalloproteinases (MMPs) are not essential for leukocyte migration across the endothelial basement membrane [41], they are critical for crossing the parenchymal border [10,58]. MMP-2 and MMP-9 can cleave β-dystroglycan, an astrocytic endfoot receptor adjacent to the parenchymal basement membrane, potentially increasing permeability [8,58]. These enzymes also selectively promote chemokine expression and astrocyte activation [10,59,60], further facilitating immune cell entry into the CNS parenchyma. Consequently, reduced MMP-9 activation in GW2580-treated animals suggests that leukocyte penetration of the parenchymal border is impaired.

We acknowledge some limitations of our study. For example, while the molecular and histological data strongly support GW2580′s effects on the BBB, we did not perform additional microscopy-based analyses of demyelination or axonal damage, or behavioral assays beyond clinical scoring. Furthermore, while our findings are consistent with a CSF1R-mediated mechanism, GW2580 can inhibit other related kinases, albeit with lower affinity [26]. Genetic approaches, such as CSF1R knockdown or knockout models, could further validate the specificity of the observed effects.

## 5. Conclusions

The results of this study demonstrate that GW2580, a selective CSF1R inhibitor, significantly reduces the severity of EAE and prevents disease relapse. GW2580′s effects are associated with reduced microglial activation, decreased leukocyte infiltration into the CNS parenchyma, and preservation of BBB integrity.

Indeed, GW2580 modulates the progression of EAE by inhibiting key pathological events. In the naïve condition, the perivascular space remains clear of immune cells, while early EAE is marked by the accumulation of peripheral immune cells in the perivascular space and increased microglial activation, indicated by elevated CSF1R expression. At the peak of EAE, immune cells breach the CNS parenchyma through astrocyte endfeet, contributing to severe inflammation. Treatment with GW2580 effectively mitigates these effects by reducing immune cell infiltration, suppressing microgliosis, and lowering the activity of matrix metalloproteinases (MMP-2/9). These actions of GW2580 correlate with a reduced expression of adhesion molecules and inflammatory mediators, ultimately leading to reduced clinical symptoms and preventing relapse.

The results from the present study, together with the reduction effect we have previously shown of GW2580 on microglial activation [33] and its safety [26,61], suggest that GW2580 could qualify as an effective, alternative remedy in MS therapy. By preserving BBB integrity and preventing immune cell migration into the CNS, GW2580 may not only alleviate acute MS symptoms but also offer long-term benefits in preventing disease progression and relapse. Future studies should focus on the long-term efficacy of GW2580 and its potential clinical applications in human MS.

## Figures and Tables

**Figure 1 cells-14-00414-f001:**
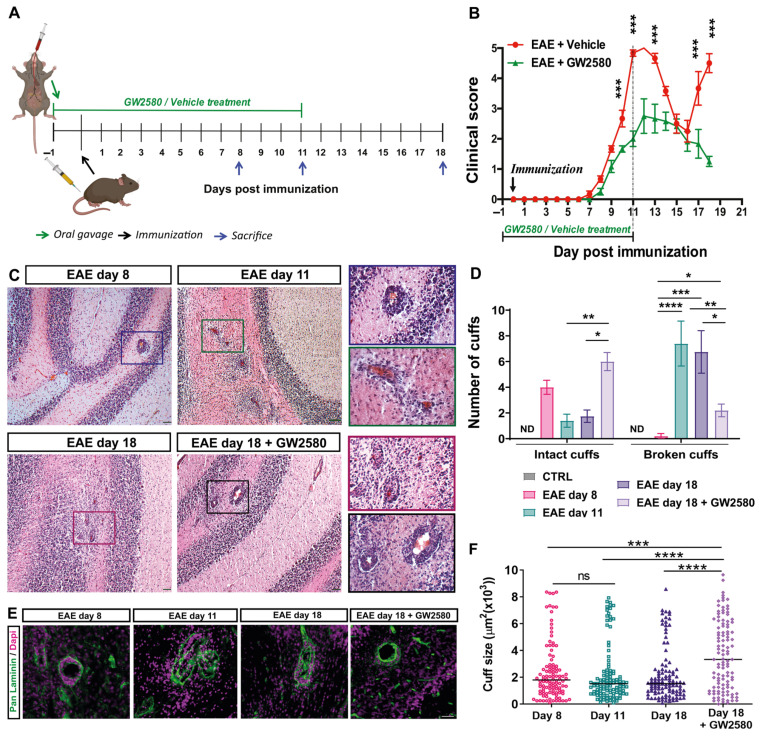
CSF1R inhibition decreased EAE clinical severity and the number of broken inflammatory perivascular cuffs. (**A**) A schematic representation of the experiment timeline. (**B**) Time course of the neurological disability score of EAE and EAE+GW2580 (means ± SD), showing EAE onset at day 8, peak at day 11 (acute phase), the remission phase at day 16, and relapse at day 18 for untreated animals and a delayed and reduced clinical score in the EAE+GW2580 group. (**C**) Hematoxylin and eosin (H&E) staining of EAE rat cerebellums at the onset of clinical signs (EAE day 8), EAE day 11: peak severity of clinical signs, EAE day 18: second peak, and EAE day 18+GW2580, showing inflammatory perivascular cuffs. Insets show higher magnification images of perivascular cuffs. (**D**) Quantification of the number of intact and broken perivascular cuffs in the healthy control and EAE rats within the different groups. (**E**) Immunofluorescence staining for pan-laminin (green) and cell nuclei (DAPI, magenta) of representative perivascular cuffs in the different EAE groups. GW2580 decreased leukocyte penetration of the perivascular cuffs. (**F**) Stereological analyses of EAE rats for cuff sizes, showing enlarged cuffs in rats treated with GW2580. Scale bars: C: 100 µm, E: 40 µm. Statistical analysis: (**B**) N = 6/group, two-way ANOVA and Bonferroni post-test (*** *p* < 0.001), (**D**) 3 images/rat (N = 6/group) were analyzed, two-way ANOVA and Tukey’s multiple-comparisons test (* *p*  <  0.05, ** *p*  <  0.01, *** *p*  <  0.001, **** *p*  <  0.0001), (**F**) cuff areas are all values measured N = 3/group, one-way ANOVA and Tukey’s multiple-comparisons test (*** *p*  <  0.001, **** *p*  <  0.0001).

**Figure 2 cells-14-00414-f002:**
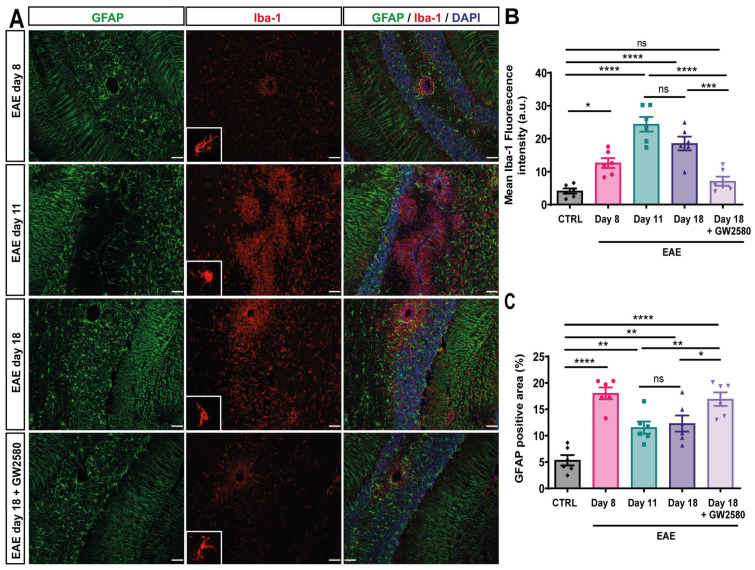
GW2580 treatment decreased glial activation in the cerebellum. (**A**) Immunofluorescence staining of thin (10 μm) cerebellum sections of EAE rats at disease onset: EAE day 8, EAE day 11: peak, EAE day 18: second peak, and EAE day 18+GW2580. Sections were stained with anti-Iba1 antibody to visualize microglia/macrophages (red), anti-GFAP antibody to mark astrocytes (green), and DAPI to visualize cell nuclei (blue). Mean Iba-1 fluorescence intensity (**B**) and the percent of GFAP-positive area (**C**) were measured in the different groups. Scale bar: 100 µm. Statistical analysis: 3 images/rat (N = 6/group) were analyzed, two-way ANOVA and Tukey’s multiple comparisons test (* *p*  <  0.05, ** *p*  <  0.01, *** *p*  <  0.001, **** *p*  <  0.0001).

**Figure 3 cells-14-00414-f003:**
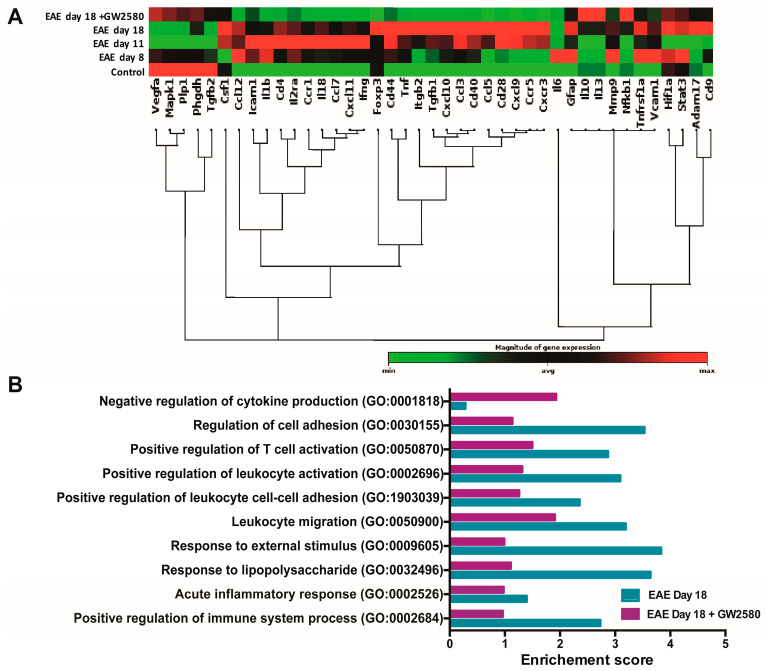
GW2580 treatment protects BBB integrity. (**A**) Clustergram of mRNA expression levels of BBB-modifying genes in the SC at different stages of EAE and with or without GW2580 treatment. Data shown are based on heat map with dendrograms, indicating the co-regulated genes across groups. Red indicates expression above the median, and green indicates expression below the median. Statistical analysis was performed using Student’s *t*-test of the replicate 2^−ΔCt^ values for each gene in the control group and treatment group; *p* < 0.05 was considered significant. (**B**) Gene ontology (GO) analysis of rats at the relapse phase of 18 DPI (EAE day 18) and under the treatment of CSF1R inhibitor GW2580 at 18 DPI (EAE day 18+GW2580). The significant terms and pathways were selected with the threshold of an adjusted *p*-value < 0.05 computed with the Fisher exact test.

**Figure 4 cells-14-00414-f004:**
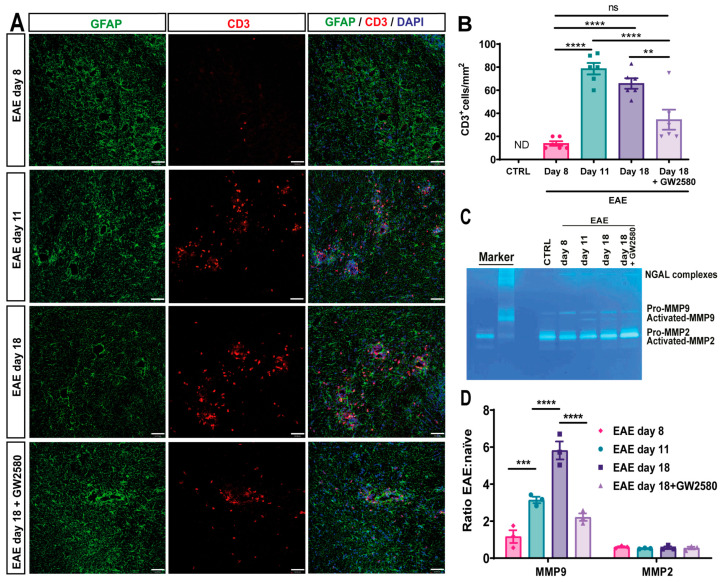
T-cell infiltration into EAE and the effects of CSF1R inhibition on MMP2/9 activation. (**A**) Representative images of 10 μm cerebellum sections immunofluorescently stained with anti-CD3 antibody to mark CD3+-infiltrating T-cells (red) and anti-GFAP antibody to mark astrocytes (green). DAPI stains all nuclei (blue). (**B**) Number of CD3+ cells/mm^2^ in control and EAE rats at different time points and under the treatment of the CSF1R inhibitor at 18 DPI (EAE day 18+GW2580). (**C**) Gelatin gel zymography of cerebellum lysates from control rats, EAE day 8, day 11, day 18, and EAE rats treated with the CSF1R inhibitor GW2580 at 18 DPI (EAE day 18+GW2580). Bands indicate pro-MMP-2 (~72 kDa), activated MMP-2 (~62 kDa), pro-MMP-9 (~92 kDa), and activated MMP-9 (~82 kDa). Human recombinant MMP-9 and mouse recombinant MMP-2 were used as markers. (**D**) Quantification of MMP-2 and MMP-9 activity, expressed as the ratio of activated to pro-enzyme forms, normalized to naive controls. GW2580 treatment significantly reduced MMP-9 activation compared to untreated EAE rats at day 18, correlating with reduced CD3+ T-cell infiltration into the CNS parenchyma. Statistical analysis: (**B**) 3 images/rat (N = 6/group) were analyzed, (**D**) quantification was performed on three different gels (N = 3 rats/group). Two-way ANOVA and Tukey’s multiple-comparisons test (** *p*  <  0.01, *** *p*  <  0.001, **** *p*  <  0.0001).

## Data Availability

Data available on request.

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
