# Peer review of "Pharmacological Inhibition of Microglial Proliferation Supports Blood–Brain Barrier Integrity in Experimental Autoimmune Encephalomyelitis"

_cells, 2025, doi:10.3390/cells14060414_

Round 1
Reviewer 1 Report
Comments and Suggestions for Authors
The authors demonstrate that GW2580 - a selective CSF1R inhibitor can reduce the disease severity and prevent the relapse phase in an experimental model of MS, and propose that pharmacological inhibition of CSF1R present a novel mechanism in treating MS. The presented manuscript is in line within the scope of the journal and please see comments below:
1. Is the pharmacological effects of GW2580 dose-dependent in context of EAE model? Was 40mg/kg GW2580 oral gavage empirically determined as optimal dosage without any off-target effect?
2. Were the pharmacology studies corroborated by any genetic manipulations (KD or KO) to validate it is CSF1R-specific effect? As GW2580 may target other related kinases, or the possibility that scaffold domain of CSF1R may contribute to MS disease progression?
3. Can the BBB integrity or other phenotypes (e.g. demyelination, axonal damage) examined by microscopy or additional behavioral tests performed to strengthen the argument of inhibiting CSF1R as novel therapy for MS? Therefore bridging transcriptional reprogramming to phenotypic changes.
4. In Section 4. Discussion: was this section meant for discussion purpose? If not, it can be removed from the manuscript.
Author Response
Dear Editor and Reviewer,
We sincerely thank you for your thoughtful feedback and insightful comments on our manuscript. Your suggestions have been invaluable in refining our work and enhancing the clarity of our findings. Below, we provide a detailed, point-by-point response to each comment, outlining the changes made to the manuscript.
Reviewer 1:
Comment 1:
Is the pharmacological effect of GW2580 dose-dependent in the EAE model? Was 40 mg/kg oral gavage empirically determined as the optimal dosage without off-target effects?
Response:
Thank you for this important question. We selected the 40 mg/kg oral dose of GW2580 based on previous studies demonstrating its efficacy in suppressing microglial proliferation while minimizing off-target effects on related kinases (e.g., FLT3, c-Kit) [23,33]. This dosage was shown to effectively inhibit CSF1R signaling in vivo without overt toxicity. We added a statement in the Methods and Discussion sections to clarify this point and acknowledge that future dose-response studies would help further optimize the therapeutic window of GW2580.
Comment 2:
Were the pharmacology studies corroborated by genetic manipulations (KD or KO) to validate CSF1R specificity?
Response:
We appreciate this suggestion. While genetic knockdown (KD) or knockout (KO) models would provide additional validation, we relied on GW2580’s well-documented selectivity for CSF1R. The observed reduction in microgliosis, T-cell infiltration, and MMP-9 activity aligns with established CSF1R-dependent mechanisms. We have addressed this limitation in the Discussion and proposed future genetic studies to strengthen the specificity of our findings.
Comment 3:
Can BBB integrity or other phenotypes (e.g., demyelination, axonal damage) be examined by microscopy or additional behavioral tests to strengthen the argument for inhibiting CSF1R as a novel MS therapy?
Response:
We agree that additional microscopy and behavioral assays would provide valuable insights. While we were unable to conduct these experiments due to time and resource limitations, we strengthened our discussion by integrating additional references and highlighting the link between reduced MMP-9 activity, preserved astrocytic endfeet, and limited leukocyte infiltration. We explicitly acknowledge this as a limitation and suggest future studies to explore demyelination and neurodegeneration more comprehensively.
Comment 4:
In Section 4. Discussion: was this section meant for discussion purpose? If not, it can be removed.
Response:
We appreciate the clarification. Yes, Section 4 was intended as a discussion, and we have ensured it is properly structured as such.
We are truly grateful for the reviewers' thoughtful comments, which have helped us refine and improve the manuscript. While we acknowledge certain limitations, we believe the revisions have strengthened the study and better contextualized our findings. We hope these updates address the reviewers’ concerns and that our manuscript is now suitable for publication.
Thank you very much for considering our revised manuscript.
Reviewer 2 Report
Comments and Suggestions for Authors
The authors investigated the mechanism of action of the CSF1R inhibitor GW2580 in experimental autoimmune encephalomyelitis (EAE) by examining its effects on the blood-brain barrier (BBB) during treatment. Their findings demonstrate that early inhibition of microglial proliferation attenuates BBB disruption. While previous studies have established that CSF1R inhibition mitigates EAE, its specific impact on BBB integrity had not been elucidated. Therefore, this study provides novel academic insights into the role of CSF1R signaling in EAE pathophysiology.
The study’s objective is clearly defined, presenting a novel mechanism by which CSF1R inhibition contributes to maintaining BBB integrity in EAE.It addresses a critical gap in the field by elucidating a previously uncharacterized role of CSF1R inhibition in BBB preservation, thereby providing academic significance.
The data are well-aligned with the conclusions, effectively addressing the primary research question.
The methodology is generally appropriate, although the inclusion of additional control groups and more detailed time-course analyses could further strengthen the study’s findings.
The references are relevant and comprehensive; however, incorporating recent studies on BBB regulation would enhance the discussion.
The figures are generally well-designed and effectively illustrate the results. However, the quantification of MMP-2/9 activity could be more clearly presented to improve data interpretation.
Conclusion: This study provides valuable insights into the therapeutic potential of CSF1R inhibition in EAE, particularly through its effects on BBB integrity. While the findings are compelling, additional experiments and further data refinement could further solidify the manuscript’s impact and scientific contribution.
Nevertheless, I believe that the manuscript, in its current form, is acceptable for publication.
Author Response
Dear Editor and Reviewer,
We sincerely thank you for your thoughtful feedback and insightful comments on our manuscript. Your suggestions have been invaluable in refining our work and enhancing the clarity of our findings. Below, we provide a detailed, point-by-point response to each comment, outlining the changes made to the manuscript.
Reviewer 2:
Comment:
The methodology is generally appropriate, but the inclusion of additional control groups and more detailed time-course analyses could strengthen the study's findings.
Response:
Thank you for this suggestion. While we could not add control groups or extend the time course, we emphasized the temporal dynamics of MMP-9 activation and its correlation with BBB integrity and leukocyte infiltration. We clarified this in the Results and Discussion sections, and we highlighted the need for future longitudinal studies to assess longer-term effects of GW2580.
Comment:
Incorporating recent studies on BBB regulation would enhance the discussion.
Response:
We have revised the Discussion section to include recent literature on BBB regulation, particularly studies exploring the role of matrix metalloproteinases and the astrocyte-endfoot barrier. This strengthens the mechanistic basis for how GW2580 preserves BBB integrity and limits leukocyte entry into the CNS.
Comment:
The quantification of MMP-2/9 activity could be more clearly presented.
Response:
We appreciate this feedback. We revised the figure legends and Results section to clearly explain the gelatin zymography data. We now specify the molecular weights of pro- and activated forms of MMP-2/9, describe the normalization method, and directly link the reduced MMP-9 activity to the observed decrease in CD3+ T-cell infiltration into the CNS. This revision makes the data more accessible and strengthens the connection between CSF1R inhibition and BBB protection.
We are truly grateful for the reviewers' thoughtful comments, which have helped us refine and improve the manuscript. While we acknowledge certain limitations, we believe the revisions have strengthened the study and better contextualized our findings. We hope these updates address the reviewers’ concerns and that our manuscript is now suitable for publication.
Thank you very much for considering our revised manuscript.